# Internet Addiction among Young Adult University Students: The Complex Interplay between Family Functioning, Impulsivity, Depression, and Anxiety

**DOI:** 10.3390/ijerph17218231

**Published:** 2020-11-07

**Authors:** Eleonora Marzilli, Luca Cerniglia, Giulia Ballarotto, Silvia Cimino

**Affiliations:** 1Department of Dynamic and Clinical Psychology, Sapienza University of Rome, Via degli Apuli, 1, cap. 00185 Rome, Italy; giulia.ballarotto@uniroma1.it (G.B.); silvia.cimino@uniroma1.it (S.C.); 2Faculty of Psychology, International Telematic University Uninettuno, 00186 Roma, Italy; l.cerniglia@uninettunouniversity.net

**Keywords:** internet addiction, family functioning, impulsivity, depression, anxiety, young adulthood

## Abstract

International research has underlined that both interpersonal, self-regulation, and comorbid variables can lead to a higher risk of developing internet addiction (IA) among young adults. To date, no studies have explored the interplay between young adults’ family functioning, impulsivity, and psychopathological difficulties. In a community sample of 244 young adult university students, this study aims to assess the relationship between young adults’ IA and young adults’ gender, the perception of their family functioning, impulsivity level, and depressive and anxiety symptoms, considering the possible interplay between these variables. The presence and the severity of IA were addressed through the Internet Addiction Test (IAT). Moreover, young adults filled out self-reporting questionnaires, assessing their perception of family functioning and their impulsivity levels and psychopathological symptoms. Results showed no significant association between the youth’s gender and IA. However, moderately addicted young adults were more likely to report poorer quality of family affective involvement and higher attentional impulsivity and depressive problems than other groups. Moreover, young adults’ attentional impulsivity mediated the relationship between family affective involvement and IA. This study provides new evidence on the complex interaction between individuals and interpersonal risk factors involved in IA among young adults, with important implications for the planning of intervention treatments.

## 1. Introduction

Over the last twenty years, the diffusion of the internet has quickly increased, becoming an integral part of daily life worldwide. Adolescents and young adults are major users of this technology [1,2,3], which allows rapid and easy access to information, interpersonal communication, entertainment, and social relationships [4,5]. From a developmental point of view, young adults have to undertake a series of evolutionary tasks to make a successful transition to adulthood (e.g., the redefinition of relationships with family, peer groups, and society; the assumption of identity, autonomy, and intimacy), and some authors have posited that internet use could help these young adults to face these important challenges [6,7,8,9]. Despite the above benefits, there is evidence that some young adults, especially university students, tend to make problematic use of the internet [10,11,12] or are even addicted to the use of the worldwide web [13].

Clinicians and researchers have shown increasing attention to internet addiction (IA) in terms of an emergent disorder [14]. Epidemiological studies on IA among young adults have reported rates from 6% to 35% [15,16,17]. Although most studies have found a higher prevalence in males [18,19], other studies have reported greater rates among females [20,21]. However, to date, the conceptualization of IA is still debated, and official diagnostic criteria have not yet been identified [22]. The fifth edition of the Diagnostic and Statistical Manual of Mental Disorders [23] has only proposed one internet-related condition—Internet Gaming Disorder—defining it as “a clinically significant impairment on daily life as a result of continual gaming”. Similarly, although there are several terms to refer to IA (e.g., problematic internet use, pathological internet use, internet dependency) [24,25,26], all of them evidence specific common characteristics, including the presence of an uncontrollable use of the internet, the need to spend more and more time online, and stress and irritability when not using the internet, with important functional impairment [27]. Therefore, given the relevance of the phenomenon, it is important to conduct further research on the possible underpinning mechanisms that may lead to the onset and maintenance of IA. In accordance with the developmental psychopathology theoretical framework [28], clinical and subclinical forms of psychopathological problems (including IA) can be considered a result of a complex interplay between individual vulnerabilities (i.e., personality traits, emotional–behavioral functioning) and relational risk factors, especially within the family context. In this field, three hypothetical and theoretical models have recently been proposed, which take into consideration the variables commonly associated with IA to explain the underpinning mechanism involved [29,30]: (a) the interpersonal impairment hypothesis, which considers IA a maladaptive response to poor quality interpersonal relationships; (b) the comorbidity hypothesis, which refers to the co-occurrence of other psychological and psychopathological difficulties; (c) the dilution effect hypothesis, which relates to self-regulation problems and the need to restore psychological well-being through internet-related problematic behaviors.

Indeed, international research has widely shown the negative effect of lack of parental support and affective involvement, family conflicts, and general poor quality of the youth’s family functioning on the substance addictive behavior of young people [31]. Recently, the same associations have also been shown with IA [32,33,34], suggesting that young adults who perceive a poor quality relationship with their parents may use the internet in a problematic way to cope with the resulting distress [35,36] and to research emotional/social support from the virtual world [37]. However, these studies have focused on samples of adolescents. To the best of our knowledge, no study has explored these possible associations among university students, although it has been evidenced that the key role for young adults’ psychological well-being is played by the quality of family functioning [38,39]. Interestingly, recent evidence has suggested that living with family members may have a protective effect on the risk of IA [40].

With regard to the comorbidity between IA and other forms of psychological problems, several studies have shown that the difficulties most commonly associated with IA are depressive and anxiety symptoms [41,42], both in clinical [41] and community samples of young adults [40,43]. Some studies have suggested that the relationship between IA and other psychopathological symptoms may be complex and bidirectional [44,45], but recent evidence has shown that problematic behaviors related to the internet are more often a strategy to manage psychological sufferance resulting from other psychopathological problems [46,47]. Among self-regulation problems, significant associations between IA and personality characteristics—especially impulsivity traits [48]—were found, which, in turn, have been shown to be significant associated with psychopathological problems, including anxiety and depressive symptoms [49,50,51]. Interestingly, it has been evidenced that the poor quality of family functioning may represent a significant contributing factor for the development of both young adults’ psychiatric symptoms [52,53,54] and impulsivity problems [55], which, in turn, may play a mediating role on the relationship between the quality of family relationships and young adults’ IA [56].

Notably, a growing body of research has suggested that contrasting findings on gender differences in epidemiological studies on IA may be a reflection of gender-related differences in risk factors for IA [57]. Specifically, some studies have found that males who reported high impulsivity traits [58] and females with anxiety and/or depressive symptoms [59] have a higher risk of developing IA compared to their peers of the other sex. It has also been shown that females are generally under higher parental supervision than males [60] and that the quality of family communication may have a different effect on male and female IA [61]. Moreover, some studies have found the presence of different developmental trajectory courses of IA among males and females, with boys reporting higher IA during adolescence, but with a more rapid decline over time when compared with girls [62].

However, although it has been suggested that the influence of family functioning may vary depending on individual features (e.g., gender and psychological profiles), to date, no study has explored whether the relationship between young adults’ family functioning and IA may be mediated by young adults’ impulsivity, anxiety, and depressive problems, also considering the possible moderation role played by gender.

Based on these premises, the present study aims to assess IA among a community sample of male and female young-adult university students, considering the role played by interpersonal factors (i.e., family functioning), comorbid psychopathological symptoms (i.e., depression and anxiety), and self-regulation (i.e., impulsivity) variables. In particular, our specific objectives are to examine (1) possible differences between male and female young adults in IA, hypothesizing a greater severity of IA among males; (2) possible association between IA, family functioning, impulsivity, and depressive and anxiety problems, hypothesizing a lower quality of family functioning and higher levels of impulsiveness and psychopathological symptoms among young adults with IA; (3) possible parallel mediation role played by young adults’ impulsivity and psychopathological symptoms on the relationship between young adults’ family functioning and IA, hypothesizing an indirect effect of affective involvement on IA through impulsivity and psychological problems.

## 2. Materials and Methods

### 2.1. Subjects Recruitment and Procedure

In collaboration with the public universities of central–south Italy, we recruited N = 350 young adults aged from 19 to 25 years. All young adults signed an informed consent form, in which the study was illustrated in detail. The study was approved by the Ethical Committee of the Department of Dynamic and Clinical Psychology at Sapienza University of Rome (protocol N. 142/2019), in accordance with the Declaration of Helsinki. All young adults who agreed to participate in the study were administered an ad hoc questionnaire to collect their sociodemographic information and a self-reporting questionnaire for the assessment of the variable under study (described below). The questionnaires were administered by expert psychologists, in randomized administration order, to small groups of university students at places made available by the universities.

### 2.2. Measures

The Internet Addiction Test (IAT) [63] is a 20-item 5-point Likert scale that measures the severity of compulsive use of the internet. Total internet addiction scores are calculated, with a maximum score of 100. According to Italian validation [64], total IAT from 0 to 30 represent average users with complete control of their internet use; scores from 31 to 49 represent the presence of a mild level of IA; scores from 50 to 79 represent moderately addicted users; scores from 80 to 100 represent severely addicted users. The scale showed very good internal consistency, with Cronbach’s alpha = 0.82 in this study.

The Family Assessment Device (FAD) [65,66], a self-reporting questionnaire, is composed of 60 items that describe various aspects of family functioning and is divided into six dimensions: Problem Solving (e.g., “We resolve most emotional upsets that come up”), Communication (e.g., “We are frank with each other”), Roles (e.g., “We discuss who is to do household jobs”), Affective Responsiveness (e.g., “We do not show our love for each other”), Affective Involvement (e.g., “If someone is in trouble, the others become involved too”), Behavioral Control (e.g., “We have rules about hitting people”). The young adult attests to the degree of concordance or discordance to which each statement describes their family. Items are scored on a 4-point Likert scale: Strongly Agree (1), Agree (2), Disagree (3), and Strongly Disagree (4). Higher scores indicate worse levels of family functioning. Psychometric properties showed good internal consistency, with Cronbach’s alpha ranging from 0.72 to 0.92 (Problem Solving, alpha = 0.77; Communication, alpha = 0.79; Roles, alpha = 0.77; Affective Responsiveness, alpha = 0.81; Affective Involvement, alpha = 0.92; Behavioral Control, alpha = 0.72).

The Adult Self Report (ASR) [67] is a self-reporting questionnaire used to get information about emotional–behavioral functioning in adults aged between 18 and 59 years. Items are assessed on a three-point Likert scale (0 = not true, 1 = somewhat true or sometimes true, and 2 = very often or very true). For the current study, we examined the ASR DSM-oriented scales, consisting of items that experts from many cultures have identified as being very consistent with DSM-5 categories. Specifically, this study aims to explore the role played by the youth’s depressive and anxiety problems on IA. Consequently, we used the scores of Depressive Problems and Anxiety Problems of the DSM-oriented scales. Previous studies have shown good reliability and validity for the ASR scales (their Cronbach’s alphas were ranged from 0.75 to 0.80) [67]. In the present study, Cronbach’s alpha for Depressive Problems was 0.77, and for Anxiety Problems, it was 0.79.

The Barratt Impulsiveness Scale (BIS-11) [68] is a 30-item self-assessment questionnaire that describes impulsive or nonimpulsive behaviors. Respondents choose from four possible alternatives along a 4-point Likert scale: 1 = Rarely/Never; 2 = Occasionally; 3 = Often; 4 = Almost Always/Always. The BIS-11 provides a total score and assesses impulsivity on the subscales of Attentional Impulsivity (inability to focus attention or concentrate, e.g., “Am a steady thinker”), Motor Impulsivity (acting without thinking, e.g., “spend more than earn”), and Nonplanning Impulsivity (lack of future orientation or forethought, e.g., “plan for job security”). The Italian version [69] showed good psychometric qualities (Cronbach’s alpha = 0.79 and test–retest reliability r = 0.889). In the present study, reliability of the three BIS-11 scales was as follows: Attentional Impulsivity = 0.78, Motor Impulsivity = 0.86, and Nonplanning Impulsivity = 0.87.

### 2.3. Statistical Analyses

Preliminary statistical analyses were carried out using descriptive statistics (reliability of the measures, frequencies, mean scores, and percentages). The IAT cutoff of Italian validation [64] was used to divide the total sample into groups (i.e., normative users, mildly addicted, moderately addicted, and severely addicted). To examine the possible association between young adults’ gender and IA, chi-square analysis was used. Then, to verify the possible associations between IA with family functioning, impulsivity, and depressive and anxiety symptoms, considering the possible role played by the youth’s gender and age, multinomial logistic regression was carried out. The three groups of IA were considered as a three-level outcome, whereas gender, age, and all considered dimensions of FAD, BIS-11, and ASR were considered as independent variables. The moderately addicted group was used as the reference category. To examine gender and age moderations, we created a series of interaction terms for inclusion in the final model, and a stepwise selection procedure was used. Consequently, only significant interaction terms appeared in the results. Model fit was assessed using the likelihood ratio chi-square test, and the results are presented in terms of odds ratio (OR), 95% CI, and *p*-value. Finally, on the total sample, to verify whether young adults’ impulsivity and psychopathological difficulties sequentially mediate the relationship between family functioning and IA, parallel mediation analyses were performed using Hayes’s [70] PROCESS macro (Model 4), which provides coefficient estimates for total, direct, and indirect effects of variables using ordinary least squares regression. Indirect (i.e., mediating) effects were evaluated with 95% bias-corrected confidence intervals based on 10,000 bootstrap samples. Confidence intervals (CIs) that do not include zero indicate effects that are significant at α = 0.05. All analyses were performed using SPSS software, version 25 [71].

## 3. Results

### 3.1. Sample Characteristics

For the aims of this study, from the total sample, we excluded young adults who did not complete the assessment procedure (*N* = 41); with mental and/or physical disability (*N* = 19); who were undergoing psychological and/or psychiatric treatment (*N* = 19); who did not accept participation in the study (*N* = 27). The final sample included 244 young adults (56.6% females) with an average age of 21.05 (SD = 2.03); 89% of the university students lived in families, and most of their parents (92%) were married or cohabiting. Based on the IAT cutoff of Italian validation [64], for Aims 1 and 2 of this study, the total sample was divided into three groups: (1) Normative Users, composed of young adults who use the internet in an adaptive way, with complete control (45/244, 18.4%); (2) Mildly Addicted, composed of young adults who spend much more time on the internet than necessary, while maintaining control of their internet use (156/244, 63.9%); (3) Moderately Addicted, composed of young adults experiencing occasional or frequent problems due to their internet use (43/244, 17.6%). In this study, no severe internet addiction risk was found among the participants. Descriptive statistics of FAD, BIS-11, and ASR by IA groups and gender are reported in Table 1.

### 3.2. Sex Differences in Internet Addiction

Chi-square analysis showed no significant association between gender and IA; χ^2^ (2, *N* = 244) = 1.43, *p* = 0.48 (Table 2).

### 3.3. Association between Internet Addiction, Family Functioning, Impulsivity, and Depressive and Anxiety Problems 

To verify possible associations between IA, demographic data, and the scores of all FAD, BIS-11, and ASR dimensions, considering the possible interaction between gender and age, and between gender and all psychosocial variables, multinomial logistic regression with gender interaction terms was carried out. Results showed a good model fit (χ^2^ = 56.86, df = 2, *p* < 0.001), but there were no significant interaction effects. However, young adults’ normative users were more likely to report lower scores of Affective Involvement of FAD (OR = 0.81, 95% CI 0.69–0.96), Attentional Impulsivity of BIS-11 (OR = 0.80, 95% CI 0.67–0.96), and Depressive Problems of ASR (OR = 0.69, 95% CI 0.59–0.80) than young adults who were moderately addicted. Moreover, mildly addicted young adults, compared to moderately addicted young adults, were more likely to report lower scores of Attentional Impulsivity (OR = 0.81, 95% CI 0.79–0.96) and Depressive Problems (OR = 0.86, 95% CI 0.77–0.95) (Table 3).

### 3.4. Young Adults’ Depressive Symptoms and Impulsivity as Mediators of the Relationship between Family Functioning and Internet Addiction

Finally, based on previous results, we verified whether impulsivity and depression mediate the effect of affective involvement perceived in the families of young adults and the score on IAT. As can be seen in Figure 1, the results of parallel mediation analyses showed that, initially, the total effect of Affective Involvement on youth IA was significant. However, considering the effects of mediators, the direct effect was reduced to a nonsignificant level. The direct effect of Affective Involvement on Attentional Impulsivity was also significant, but on Depression, it was not. The direct effects of mediating variables on IAT were also significant. Moreover, it was found that the model showed overall significant levels and explained 22% of the variance in IA.

Regarding indirect effects, Table 4 shows that the indirect paths via Attentional Impulsivity were statistically significant. Conversely, the mediation of Depression was not significant.

## 4. Discussion

This study aims to examine the complex relationship between individuals and relational risk factors (i.e., young adults’ gender, impulsivity, depressive and anxiety symptoms, and perception of their family functioning) that literature has shown to be commonly associated with IA among young-adult university students. We have chosen to explore these relationships in a sample of university students because international research has shown that many young adults suffer from subthreshold levels of IA [72,73], which may lead to maladjustment to their living environment [74]. In line with previous studies in this field, we did not find any subject with severe IA, but most of them were mildly addicted (i.e., 63.9%) and 17.6% were moderately addicted. Consequently, based on the severity of IA, we divided the sample into three groups to examine the possible risk factors associated.

With regards to possible differences related to gender in young adult IA, we hypothesized to find a greater severity among males. However, the results did not confirm what was expected, showing no significant association between the three IA groups and the youths’ gender. Several studies have suggested that males have a higher risk in most addictive behaviors [75,76], including IA [18,19], due to their generally lower self-control [77] and tendency to engage in high-risk additive internet activities (e.g., online videogames or cybersex) [78]. However, our results are in accordance with recent evidence that the gender gap in time spent on the internet is reducing [79] and that the prevalence of IA among females is increasing [20,80], with no significant gender differences [21,81,82].

In this context, some studies have shown the presence of gender-related differences in risk factors associated with IA [57], which may be responsible for different trajectory patterns of IA among males and females. Moreover, a study by Li and colleagues [62] found that although males reported higher IA scores than females at the onset of the disease, they showed a more rapid reduction of rates over time compared to females. However, our results showed no significant interactions between gender and any of the variables considered. In this regard, it is important to note that the relationship between IA and other psychopathological difficulties may be complex and bidirectional [45]. A study by Liang and colleagues [83] has shown that psychopathological symptoms (i.e., depressive symptoms) were significant predictors of IA among males, but in females, IA was a significant predictor of subsequent depressive symptoms. This evidence could explain the lack of significant gender-related effects on IA, which should be further explored in future longitudinal studies. However, the results confirmed significant associations between IA and impulsivity, psychopathological symptoms, and family functioning. In particular, moderately addicted young adults were more likely to report poorer affective involvement of their families compared to normative users. These findings are consistent with previous studies that have directly linked poor family functioning to the onset, development, and maintenance of several risk-taking and addiction-related behaviors [31,84,85], including IA [32,33,34]. As suggested by Hosseinbor and colleagues [31], families characterized by a low affective involvement, in which there is no parental investment in the young adult’s daily life issues, can affect the children’s self-control abilities. Consequently, young adults may tend to overuse the internet, both to look for useful information to solve their problems [34] and as a strategy to cope with the distress and psychological sufferance resulting from unsatisfactory relationships with parents [35,36] by obtaining emotional and social support outside their families [86,87]. Moreover, as expected, our results have shown that university students with internet overuse/abuse were more likely to report higher attentional impulsivity and depressive problems than other groups. No significant associations were found with anxiety problems and other dimensions of BIS-11 and FAD. Although previous studies have shown a significant association between IA and anxiety [42,43], our results did not confirm this relationship. A possible explanation could be the different tools used by those studies to assess anxiety and heterogeneity within the scale of anxiety problems assessed by the Adult Self Report [88]. Indeed, it includes general anxiety disorder, social anxiety disorder, and specific phobia, and recent studies have suggested that social anxiety may have a specific role in predicting IA among young adults who have social anxiety [89,90] and/or phobic anxiety [42,91]. However, our study is in line with the study by Mamun and colleagues [92], who found a significant association between IA and depression but not with anxiety. In this field, a growing body of research has shown that among psychiatric risk factors for IA, depression has the greatest size effect [41,42]. Previous studies have also shown a frequent co-occurrence between internalizing problems and high levels of impulsivity [93,94], which share common neural substrate vulnerabilities that are primarily involved in the reward system [95,96,97,98]. In this field, recent longitudinal studies have evidenced a predictive role played by attentional impulsivity and depression on young adult IA [99,100]. Young adults with higher levels of attentional impulsivity have difficulties in cognitive persistence and in focus and concentration [101], tending to experiment frequently in intrusive and irrelevant thoughts [102,103]. These characteristics may facilitate an engagement in internet-related activities, predisposing to excessive use of the internet [104] as a maladaptive way to manage with nonrelevant and intrusive thoughts [105,106,107]. On the other hand, young adults with depressive symptoms may use the internet as a maladaptive coping strategy to deal with negative feelings of sadness/desperation, apathy, and loneliness [27]. 

It is important to note that recent evidence has suggested that the nature of the association between family functioning, impulsivity traits, depressive problems, and IA can be dynamic and complex. In particular, some studies have shown that poor quality family functioning is prospectively associated with a broad range of psychological difficulties, including high impulsivity and depression symptoms [52,53,54,55], which, in turn, may lead to a higher risk for the onset of IA [41,43,48]. Consequently, based on our previous results, we tested whether these two factors (i.e., attentional impulsivity and depression) may be parallel mediators of the relationship between family functioning (i.e., the family affective involvement) and IA. The results showed that the relationship between the affective involvement of a young adult’s family and IA, via attentional impulsivity, was significant. Although the total effect of affective involvement on IA was significant, considering the effect of the mediator variables, the direct effect was not significant. Notably, the direct effect of affective involvement on depression and the indirect effect of affective involvement on IA through depression were not significant. Despite the fact that the studies by Li and colleagues [108] and by Shi and colleagues [109] have underlined that the effect of youths’ family functioning on IA could be mediated by other individual vulnerabilities (e.g., low self-esteem, social sensitivity, loneliness), to our best knowledge, this is the first study that has explored this association on a sample of young-adult university students, considering the possible mediation role played by impulsivity and depressive problems. 

This study has some limitations. First, although young adults’ IA, the quality of their family functioning, impulsivity levels, and depressive and anxiety problems were assessed through validated and widely used self-reporting tools, studies should use more robust methodologies for the evaluation of these variables (e.g., observational procedures or clinical interviews). Some individual, relational, and parental variables that literature has shown to be commonly associated with young adults’ psychopathological symptoms, impulsivity, and IA were not investigated (such as genetic vulnerabilities, the quality of peer relationships, parental psychopathological risks, marital conflicts) [12,110,111,112,113,114]. Moreover, the homogeneity of the sample in terms of sociodemographic characteristics limits the generalizability of the results to a population with limited income and education levels. Therefore, further studies should consider samples with low socioeconomic status and educational levels. Finally, the cross-sectional nature of our study implies that caution needs to be taken on the causal links found, which should be supported by future longitudinal research. Despite these limitations, although several studies have shown that poor family functioning is one of the main risk factors for IA among the adolescent population, this study was the first to explore this association among young-adult university students, also considering the complex interplay with young adults’ gender, impulsivity, and depressive problems. Our findings may suggest a key role played by youths’ impulsivity level (i.e., attentional impulsivity) on the relationship between family functioning and IA for both males and females, which may be informative for the planning of more targeted and effective intervention treatments. Moreover, in our sample, most of the young-adult university students reported the presence of internet-related problematic behaviors (from mild to moderate addictive levels), supporting the importance of implementing research on this population group to add new knowledge of the underpinning mechanisms involved in IA and to prevent the emergence of severe forms of IA.

## 5. Conclusions

A growing body of research on risk factors associated with IA among young adulthood has shown that the key to a richer understanding of the etiopathogenetic mechanisms is to pay attention to how risk factors of a different nature (from individual to relational domains) work together in a complex way. This study has provided new preliminary evidence on the implication of the family functioning of young adults and its interplay with impulsivity and psychopathological problems. In particular, moderately addicted young adults were more likely to report poorer parental affective involvement and higher attentional impulsivity and depressive problems than youths who use the internet in an adaptive way. Our findings may help to explain the process of how these variables affect IA among youths. Specifically, experiencing low affective parental involvement may lead the youth to become more attentionally impulsive, which, in turn, may lead to excessive use of the internet as a strategy to cope with the resulting psychological suffering. Although the association between poor quality family functioning and depressive problems was not significant, our results suggest that depressive symptoms may also directly increase the risk of IA as a strategy to cope with the resulting psychological suffering. Overall, our preliminary findings may provide additional preliminary support for the importance of planning family-focused prevention programs for all young adults at risk of IA. Prevention focused on early detection and intervention in impulsivity and depression also seems to be necessary for both males and females. However, further research, with larger sample sizes of young adults, is needed to generalize our preliminary findings. Moreover, given the cross-sectional nature of the study, further longitudinal studies to explore IA trajectories could be useful for identifying more accurate risk factors for IA and supporting the causal connection between young adults’ family functioning, impulsivity, depressive problems, and IA that is suggested by our preliminary findings.

## Figures and Tables

**Figure 1 ijerph-17-08231-f001:**
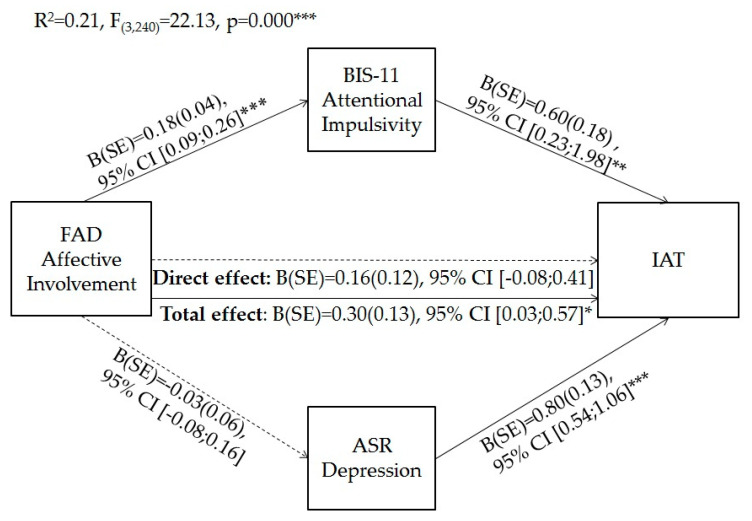
Parallel mediation of attentional impulsivity and depression in the relationship between affective involvement and internet addiction. * *p* < 0.05, ** *p* < 0.01, *** *p* < 0.001.

**Table 1 ijerph-17-08231-t001:** Descriptive statistics for mean scores by internet addiction (IA) group and gender.

			Youth’s IA		
			Normative Users M (SD)	Mildly Addicted M (SD)	Moderately Addicted M (SD)
FAD	Problem Solving	Male	12.60 (2.66)	12.78 (2.91)	12.81 (3.08)
		Female	14.36 (3.43)	13.19 (2.87)	14.38 (2.97)
		Total	13.57 (3.20)	13.02 (2.88)	13.58 (3.09)
	Communication	Male	19.40 (4.01)	19.67 (4.47)	21.22 (4.21)
		Female	22.08 (5.12)	21.66 (4.67)	22.71 (3.88)
		Total	20.88 (4.80)	20.84 (4.68)	21.95 (4.08)
	Roles	Male	23.65 (5.34)	24.73 (5.24)	25.95 (4.83)
		Female	26.56 (6.09)	26.26 (5.25)	29.33 (6.67)
		Total	25.26 (5.89)	25.69 (5.29)	27.60 (5.98)
	Affective Responsiveness	Male	14.45 (3.60)	13.92 (3.88)	15.31 (3.74)
		Female	16.24 (7.38)	15.14 (4.27)	16.71 (5.01)
		Total	15.44 (6.01)	14.64 (4.15)	16 (4.41)
	Affective Involvement	Male	15.05 (4.03)	16.07 (4.29)	17.09 (3.93)
		Female	16.28 (4.38)	17.38 (5.20)	19.71 (4.65)
		Total	15.72 (4.22)	16.84 (4.88)	18.37 (4.45)
BIS-11	Behavioral Control	Male	18.15 (4.82)	18.23 (4.41)	19.22 (4.61)
		Female	18.60 (3.50)	18.89 (4.89)	20.90 (5.30)
		Total	18.40 (4.09)	18.62 (4.69)	20.04 (4.97)
	Attentional Impulsivity	Male	16.85 (3.08)	17.59 (2.85)	18.36 (2.88)
		Female	14.28 (2.60)	16.08 (3.45)	17.71 (2.86)
		Total	15.42 (3.07)	16.70 (3.29)	18.04 (2.86)
	Motor Impulsivity	Male	10.45 (3.45)	20.39 (3.34)	20.63 (3.97)
		Female	18.28 (3.64)	19.36 (3.55)	20.66 (4.19)
		Total	19.24 (3.68)	19.94 (3.47)	20.65 (4.03)
	Nonplanning Impulsivity	Male	26.35 (5.86)	26.31 (4.56)	25.68 (4.98)
		Female	24.12 (5.65)	25.91 (4.88)	28.38 (6.43)
		Total	25.11 (5.79)	26.07 (4.74)	27 (5.83)
ASR	Depression	Male	3.85 (3.85)	6.03 (3.78)	8.32 (4.72)
		Female	5.04 (4.39)	8.16 (4.41)	11.14 (3.75)
		Total	4.51 (4.15)	7.29 (4.28)	9.70 (4.45)
	Anxiety	Male	5.15 (2.23)	5.28 (2.51)	5.45 (2.97)
		Female	5.28 (1.96)	6.14 (2.27)	6.71 (2.70)
		Total	5.22 (2.10)	5.79 (2.40)	6.07 (2.88)

Table Footer:. FAD = Family Assessment Device; BIS-11 = Barratt Impulsiveness Scale; ASR = Adult Self Report

**Table 2 ijerph-17-08231-t002:** Association between young adults’ gender and internet addiction (IA).

		Young Adults’ IA			Total
Sex		Normative Users	Mildly Addicted	Moderately Addicted	
Male	*N*	20	64	22	106
	% within IA group	44.4	41	51.2	43.4
	% within sex	18.8	60.4	20.8	
Female	*N*	25	92	21	138
	% within IA group	55.6	59	48.8	56.6
	% within sex	18.1	66.7	15.2	
Total	*N*	45	156	43	244

**Table 3 ijerph-17-08231-t003:** Multinomial logistic regression comparing normative users and mildly addicted users with moderately addicted users.

		Normative Group				Mildly Addicted			
		B (SE)	OR	95% CI	*p*	B (SE)	OR	95% CI	*p*
	Age	0.15 (0.14)	1.16	[0.88, 1.53]	0.28	0.08 (0.17)	1.08	[0.94, 1.36]	0.49
	Gender ^a^	0.79 (0.55)	1.22	[0.74, 3.16]	0.15	0.84 (0.42)	0.97	[0.86, 1.36]	0.12
	Living setup ^b^	0.65 (0.12)	0.95	[0.81, 2.23]	0.35	0.88 (0.09)	0.98	[0.75, 2.12]	0.45
FAD	PS	0.12 (0.10)	1.13	[0.92, 1.40]	0.23	0.01 (0.08)	1.01	[0.87, 1.19]	0.82
	Com	0.05 (0.08)	1.05	[0.88, 1.25]	0.53	0.04 (0.06)	1.04	[0.91, 1.19]	0.49
	Rol	0.04 (0.07)	1.04	[0.90, 1.20]	0.55	−0.01 (0.05)	0.99	[0.89, 1.10]	0.90
	AffRes	−0.20 (0.08)	1.04	[0.89, 1.22]	0.59	−0.04 (0.06)	0.95	[0.84, 1.09]	0.52
	AffInv	−0.03 (0.08)	0.81	[0.69, 0.96]	0.01 **	−0.04 (−0.06)	0.96	[0.85, 1.07]	0.49
	BC	0.10 (0.07)	0.96	[0.83, 1.12]	0.65	−0.03 (0.05)	0.96	[0.86, 1.08]	0.55
BIS−11	AttImp	−0.21 (0.09)	0.80	[0.67, 0.96]	0.01 **	−0.10 (0.06)	0.81	[0.79, 0.96]	0.01 **
	MotImp	0.06 (0.08)	1.06	[0.90, 1.25]	0.42	0.02 (0.05)	1.02	[0.91, 1.14]	0.65
	NPlan	0.04 (0.05)	1.04	[0.93, 1.16]	0.45	0.02 (0.04)	1.02	[0.94, 1.10]	0.60
ASR	Dep	−0.36 (0.07)	0.69	[0.59, 0.80]	0.001 ***	−0.15 (0.05)	0.86	[0.77, 0.95]	0.04 *
	Anx	0.10 (0.12)	1.11	[0.87, 1.41]	0.37	0.09 (0.09)	1.10	[0.91, 1.31]	0.30

Note: ^a^ Female is the reference group; ^b^ Living with family members is the reference group; SE = standard error, OR = odds ratio, CI = confidence interval; FAD = family assessment device; BIS-11 = Barratt Impulsiveness Scale; ASR = Adult Self Report; PS = problem solving; Com = communication; Rol = roles; AffRes = affective responsiveness; AffInv = affective involvement; BC = behavioral control; AttImp= attentional impulsivity; MotImp = motor impulsivity; NPlan = nonplanning impulsivity; Dep = depressive problems; Anx = anxiety problems. * *p* < 0.05, ** *p* < 0.01, *** *p* < 0.001.

**Table 4 ijerph-17-08231-t004:** Indirect effects of affective involvement on internet addiction through attentional impulsivity and depression.

Indirect Effect	Effect (BootSE)	LLCI	ULCI
Total	0.14 (0.06)	**0.01**	**0.28**
Affective Involvement → Attentional Impulsivity → IA	0.11 (0.04)	**0.03**	**0.20**
Affective Involvement → Depression → IA	0.03 (0.04)	−0.06	0.13

Note: BootSE = boot-strapped standard error; LLCI = lower level confidence interval; ULCI = upper level confidence interval. All bold values are statistically significant (CI did not contain zero).

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
