# Peer review of "Internet Addiction among Young Adult University Students: The Complex Interplay between Family Functioning, Impulsivity, Depression, and Anxiety"

_ijerph, 2020, doi:10.3390/ijerph17218231_

Round 1

Reviewer 1 Report

Thank you for the opportunity to review this very interesting and well-written paper examining the relationship between internet addiction, family functioning, impulsivity, depression, and anxiety. I enjoyed reading the paper and feel it would make a welcome contribution to the field. That said, I do have some methodological/theoretical concerns, and concerns regarding language, specifically using tentative language. I note my specific comments below.

  1. Overall, the manuscript was very well-written. However, there are odd sentences and phrasing here and there. I would suggest a further review to strengthen the paper. For example, level of IA in abstract could be changed to severity of IA. I am also unclear as to the sentence "Internet are also functional to face a series of 37 evolutionary tasks specific to emerging adulthood" (page 1 lines 37-39). Page 2, lines 53-54, rather than implement the research, conduct research? As regard to, In regards to? There are further examples throughout.
  2. The authors use the term general population throughout the paper to describe their sample. However, it seems participants were university students. If so, I suggest changing general population to university students throughout the paper, including in the title.
  3. The authors use the term young adults in the title and youths in the paper. I would suggest keeping it consistent throughout the paper. The term young adults seem most appropriate given the age ranges of 19-25
  4. Throughout the manuscript, I would suggest using more tentative language about the results. For example, only one facet of family functioning, and impulsivity was significant. In other words, more facets were non-significant. Furthermore, if you control for multiple comparison on Table 2, the results would not be significant. As such, I would say the results provide preliminary evidence and MAY suggest X, Y, Z
  5. When reading the manuscript, I thought the rationale and literature review would suggest a parallel mediation model, not a sequential mediation model. Although on page 2 the authors state that "Among youth’s self-regulation problems, significant associations between IA and 77 personality characteristics - especially impulsivity traits [44] - were found, which in turn have been 78 shown to be significant predictors of psychopathological problems, including anxiety and 79 depressive symptoms [45-47].", the above references are all cross-sectional and do not specify causation and citation 47 may even suggest anxiety --> impulsivity direction. As such, the introduction needs much stronger evidence to suggest the sequential mediation model for impulsivity --> depression. Otherwise, I would suggest running a parallel mediation model.
  6. An aim of the present study according to the authors is gender differences page 2 lines 93-93. Yet, the intro is significantly lacking in background for this, other than prevalence rates. The authors do a fantastic job regarding the mediation model. Thus, I would suggest more space in the introduction be given to gender differences, especially in light of findings that the relationship between IA and psychopathology may be dependent on gender e.g. Liang, L., Zhou, D., Yuan, C., Shao, A., & Bian, Y. (2016). Gender differences in the relationship between internet addiction and depression: A cross-lagged study in Chinese adolescents. Computers in Human Behavior, 63, 463-470.
  7. It would also be informative if the authors provided a supplementary table regarding gender differences in family functioning, impulsivity, depression and anxiety. Could these differences account for the null result in IAT?
  8. In the measures, with the exception of the IAT whether the alphas were from the present sample. I would suggest the authors including the alpha from the sample itself.
  9. In the statistical analyses, I am unclear as to what is the qualitative analysis for this study. Descriptives are still qualitative, and thus I would suggest removing the word qualitative.
  10. In results for sample characteristics, I would make it clearer that no participant fell into the severe category. This is not specified until the discussion, which made it confusing as to the three groups.
  11. Table 1, it is more informative to provide % for chi-square tables rather than Exp. Val, and St.R.
  12. In the discussion, it may be helpful to discuss why IAT was not associated with anxiety in the sample, despite previous studies suggesting a relationship between the two
  13. I feel the discussion would be strengthened if the authors could speak to some of the potential clinical significance, specifically regarding prevention and potential treatment. These should be framed as preliminary and balanced with the findings. 

Author Response

Response to Reviewer 1 Comments

Point 1: Overall, the manuscript was very well-written. However, there are odd sentences and phrasing here and there. I would suggest a further review to strengthen the paper. For example, level of IA in abstract could be changed to severity of IA. I am also unclear as to the sentence "Internet are also functional to face a series of 37 evolutionary tasks specific to emerging adulthood" (page 1 lines 37-39). Page 2, lines 53-54, rather than implement the research, conduct research? As regard to, In regards to? There are further examples throughout.

Response 1: We thank the reviewer for these comments. We have changed the term “level of IA” to “severity of IA”, both in the abstract and throughout the manuscript (See lines 19, 112, 263, and 266). Moreover, we have modified the sentence underlined by the reviewer (See lines 36-40, and 55) and other sentences throughout the manuscript (See lines 60-62, 79, and 265)

Point 2: The authors use the term general population throughout the paper to describe their sample. However, it seems participants were university students. If so, I suggest changing general population to university students throughout the paper, including in the title.

Response 2. We thank the reviewer for this pertinent comment. As suggested by the reviewer, we have modified the title in “Internet addiction among young adult university students: the complex interplay between family functioning, impulsivity, depression and anxiety” and described our sample in terms of university students (See lines 16,75,108, 129, 200, 295,334,350, and 354)

Point 3: The authors use the term young adults in the title and youths in the paper. I would suggest keeping it consistent throughout the paper. The term young adults seem most appropriate given the age ranges of 19-25

Response 3. As suggested by the reviewer, we have used the most appropriate term (ie., young adults) both in the title, in the abstract, and throughout the manuscript (See, for example, lines 41,71,90, 92, 108, 11, 115, 121, 122, 125, 176, 186)

Point 4. Throughout the manuscript, I would suggest using more tentative language about the results. For example, only one facet of family functioning, and impulsivity was significant. In other words, more facets were non-significant. Furthermore, if you control for multiple comparison on Table 2, the results would not be significant. As such, I would say the results provide preliminary evidence and MAY suggest X, Y, Z

Response 4. We thank the reviewer for this pertinent comment. We have further highlighted that ours are preliminary results that may suggest the presence of causal relationships that have emerged (see lines 351-354, and the Conclusion section)

Point 5. When reading the manuscript, I thought the rationale and literature review would suggest a parallel mediation model, not a sequential mediation model. Although on page 2 the authors state that "Among youth’s self-regulation problems, significant associations between IA and 77 personality characteristics - especially impulsivity traits [44] - were found, which in turn have been 78 shown to be significant predictors of psychopathological problems, including anxiety and 79 depressive symptoms [45-47].", the above references are all cross-sectional and do not specify causation and citation 47 may even suggest anxiety --> impulsivity direction. As such, the introduction needs much stronger evidence to suggest the sequential mediation model for impulsivity --> depression. Otherwise, I would suggest running a parallel mediation model.

Response 5. We thank the reviewer for this useful comment. As the reviewer has evidenced, the current literature does not support the existence of a possible sequential mediation model, given the cross-sectional nature of studies focused on the relationship between impulsivity and depression. Consequently, as suggested by the reviewer, we have modified the relative sentences throughout the manuscript and our aim (see line 115),  to verify the possible presence of a parallel mediation model. Coherently, we have modified the statistical analyses subsection (see line 188), the results section (lines 237-253), the Figure showing results of parallel mediation model (see new Figure 1), the discussion of the relative results (See line 323-330), and the conclusions of the study (see lines 368-373).

Point 6. An aim of the present study according to the authors is gender differences page 2 lines 93-93. Yet, the intro is significantly lacking in background for this, other than prevalence rates. The authors do a fantastic job regarding the mediation model. Thus, I would suggest more space in the introduction be given to gender differences, especially in light of findings that the relationship between IA and psychopathology may be dependent on gender e.g. Liang, L., Zhou, D., Yuan, C., Shao, A., & Bian, Y. (2016). Gender differences in the relationship between internet addiction and depression: A cross-lagged study in Chinese adolescents. Computers in Human Behavior, 63, 463-470.

Response 6. We thank the reviewer for this comment. We have added more information about the role played by sex and sex-related differences in the variables under study, which may have a different role in predicting the risk of IA among males and females (see lines 93-101, of the Introduction, and lines 273-283 of the Discussion). Consequently, to verify the possible moderation role played by young adults’ sex on the relationship between psychosocial variables and IA, we have conducted a multinomial logistic regression including a series of interaction terms between sex and the other variables (see lines 177-186 of the statistical analyses, lines 217-226 of the results, the new Table 3, and lines 285,286, 295-298 of the Discussion)

Point 7. It would also be informative if the authors provided a supplementary table regarding gender differences in family functioning, impulsivity, depression and anxiety. Could these differences account for the null result in IAT?

Response 7. We thank the reviewer for this useful comment. In order to verify whether possible sex differences in family functioning, impulsivity, depression, and anxiety may account for the relationship between these risk factors and IA, as explained above, we have chosen to conduct a multinomial logistic regression, including a series of interaction terms between sex and the other variables (see lines 177-186 of the statistical analyses, lines 217-226 of the results, the new Table 3, and lines 285,286, 295-298 of the Discussion). Preliminary, we have reported descriptive mean of psychosocial variables, both by IA-group and sex (See new Table 1, page 5).

Point 8. In the measures, with the exception of the IAT whether the alphas were from the present sample. I would suggest the authors including the alpha from the sample itself.

Response 8. As suggested by the reviewer, we have added the alphas for all instruments (See lines 148-150, 160, 169, and 170).

Point 9. In the statistical analyses, I am unclear as to what is the qualitative analysis for this study. Descriptives are still qualitative, and thus I would suggest removing the word qualitative.

Response 9. As suggested by the reviewer, we have removed the word qualitative in relation to statistical analyses conducted.

Point 10. In results for sample characteristics, I would make it clearer that no participant fell into the severe category. This is not specified until the discussion, which made it confusing as to the three groups.

Response 10. We thank the reviewer for this pertinent comment. We have explained from the description of results that we did not find any young adults into the severity category (See line 207)  

Point 11. Table 1, it is more informative to provide % for chi-square tables rather than Exp. Val, and St.R.

Response 11. As suggested by the reviewer, we have reported the % within sex and within IA-groups in the new Table 2, page 214.

Point 12. In the discussion, it may be helpful to discuss why IAT was not associated with anxiety in the sample, despite previous studies suggesting a relationship between the two

Response 12. As suggested by the reviewer, we have added a possible explanation for the lack of association emerged between anxiety and IA. This could be due to the assessment tool used in our study (ie, ASR), which included a heterogeneity of anxiety problems (ie., General Anxiety Disorder, Social Anxiety Disorder, and Specific Phobia), and recent studies have suggested that social anxiety [89,90] and/or phobic anxiety [42,91] may have a more specific role in predicting IA among young adults.

Point 13. I feel the discussion would be strengthened if the authors could speak to some of the potential clinical significance, specifically regarding prevention and potential treatment. These should be framed as preliminary and balanced with the findings. 

Response 13. As suggested by the reviewer, we have added potential clinical implication of our study. Specifically, our findings may suggest a key role played by youths’ impulsivity level (i.e., attentional impulsivity) in the relationship between family functioning and IA, for both males and females, providing additional preliminary support for the importance of the planning of family-focused prevention programs for all young adults at risk of IA. Moreover, based on our preliminary findings, prevention focused on early detection and intervention in impulsivity and depression problems seems to be also necessary, for both males and females (See lines 351-354 of the Discussion, and lines 373-376 of the Conclusion)

Thank you again for your kind attention.

Best regards

The authors.

Reviewer 2 Report

This manuscript addresses the role of internet addiction in a complex interplay among youths. Generally, I appreciate author’s efforts on comprehensive literature review and statistic analysis. However, I have several concerns listed as follows:  

  1. In the end of Introduction, I suggest to add some descriptions about the gap of literatures that can be covered by your study.
  2. In the line 159, do you mean the alpha level at “0.05”?
  3. I am wondering that why you divided the populations into three group, but not directly analyzed it with linear regression (Dependent variables: scores of IA; independent variables: other clinical variables and demographic information).
  4. Regarding the analysis for demographic data, only sex is analyzed. Is there any other information collecting in this study, such as age, educational level, marital status, etc.
  5. In the table 2, the definition of “a, b, c” are missing. If you are intended to address it in the supplementary table, I suggest add brief description in the footnotes.
  6. I am really interested in the conceptual model. Since “impulsivity” can be directed associated with “IAT”, what is the clinical importance of “depression”? In addition, have you ever tried other Hayes's model to test the association between variables, such as model 14?

Author Response

Response to Reviewer 2 Comments

Point 1. In the end of Introduction, I suggest to add some descriptions about the gap of literatures that can be covered by your study.

Response 1. We thank the reviewer for this pertinent comment. As suggested by the reviewer we have made more explicit the gap of literatures that can be covered by our study (See lines 74-78, 102-106)

Point 2. In the line 159, do you mean the alpha level at “0.05”?

Response 2. As suggested by the reviewer, we have modify the typing error (see line 192)

Point 3. I am wondering that why you divided the populations into three group, but not directly analyzed it with linear regression (Dependent variables: scores of IA; independent variables: other clinical variables and demographic information).

 Response 3. We thank the reviewer for this pertinent comment. As suggested by the reviewer, we have modified our statistical analyses. Specifically, after conducting descriptive analyses and chi-square analyses, we have directly analyzed the possible relationship between psychosocial and demographical variables and IA group with multinomial regression analyses. We have considered as dependent variable the three-group of IA whereas sex, age, living setup, and all considered dimension of FAD, BIS-11, and ASR as independent variables. Moreover, to examine possible sex and age moderations, we created a series of interaction terms for inclusion in the final model. (see lines 177-186 of the statistical analyses, lines 217-226 of the results, the new Table 3, and lines 285,286, 295-298 of the Discussion).

Point 4. Regarding the analysis for demographic data, only sex is analyzed. Is there any other information collecting in this study, such as age, educational level, marital status, etc.

Response 4. As suggested by the reviewer, and as explained above, we have carried out a multinomial logistic regression, also considering the possible role played by sex, age, and living setup. We have chosen to consider these demographical variables because previous studies have suggested that they may have a role in the risk of IA, among young adults population (See lines 77-78, 93-101).

Point 5. In the table 2, the definition of “a, b, c” are missing. If you are intended to address it in the supplementary table, I suggest add brief description in the footnotes.

Response 5. We thank the reviewer for this comment. However, as explained above, we have modified our statistical analyses, and MANOVA has not been conducted anymore. However, we have reported descriptive mean of psychosocial variables, both by IA-group and sex (See new Table 1, page 5).

Point 6. I am really interested in the conceptual model. Since “impulsivity” can be directed associated with “IAT”, what is the clinical importance of “depression”? In addition, have you ever tried other Hayes's model to test the association between variables, such as model 14?

Response 6. We thank the reviewer for this comment. From a more accurate examination of the literature we thought it more useful to redefine our conceptual model. In fact, studies in the field, rather than suggesting a predictive effect of depression on impulsivity, show the presence of associations and are mostly correlational studies. In relation to the possible predictive role played by impulsivity on depression, previous studies have shown the specific role played by motor impulsivity, but not attentional impulsivity (see, for example, the reference [50]: Regan, T., Harris, B., Fields, S.A. Are relationships between impulsivity and depressive symptoms in adolescents sex-dependent?. Heliyon. 2019, 5, e02696. doi: 10.1016/j.heliyon.2019.e02696).

In light of these reflections, and on the basis of the previous literature that has highlighted the specific role played by impulsivity and depression, separately, on IA, we considered more appropriate to test a parallel mediation model, using model 4. Moreover, to implement our analysis plan, as highlighted above, we tested the possible sex moderation on the relationship between psychosocial variables and IA (as partially suggested by model 14) with multinomial regression analyses. Given that results showed no significant interaction effects, we subsequently tested models of non-moderated mediation (ie, the model 4).

Thank you again for your kind attention.

Best regards

The authors.

Reviewer 3 Report

Review of IJERPH

 In general, this was a straightforward paper to review.  I am concerned about the results section.  The authors categorize respondents into categories based on the “IAT” but then treat the scale as continuous for their models.  The two methods should not be done in the same paper.  Tables 1 and 2 should be a single correlation matrix with age and sex included within the matrix.  The authors also do a very poor job in explaining their “serial and separate mediation” models.  A single straightforward regression predicting internet addiction would have been much better for this manuscript.

On page 3:

- make sure you use quotation marks around the test item examples

- explain what is meant when you say, “each family member attests...”.  The text above suggests that only students completed the scale and not their family members 

- specify which scale had an alpha of .72 and which was .92 (line 125)

- state the alpha value for the ASR for this sample

- state the alpha value for the BIS-11 for this sample

On page 4, note that reliability values are not qualitative as claimed on line 146, but are indeed quantitative

In addition, the authors claim on page 6, line 222, that their sample is from “the general population”.  This claim is incorrect as the sample consists of university students.

More minor points:

- insert the reference for the IAT in the Abstract number 54.

- page 2 – “at this regard” is an awkward phrase and the rest of the sentence needs rewriting for clarity

- line 53, remove “the” before “research”

- lines 65 and 72, change “showed” to “shown”

- line 69, remove “the” before “relationship”

- line 72, you need to add what variable was “associated”; associated with....IA? finish the clause

- line 85, insert “the” before “influence”

- line 110, change to “maximum”

- line 111, change to “score” (singular)

- line 120, clarify that the item is negatively keyed (“We don’t...”)

- line 147, change to read “was used”

- line 163, should be “did not complete”

- line 167, delete “The” and write out the number

- line 168, delete “are” before “married”

- line 173, explicitly state that none of the sample met the criteria of “severely addicted”

- line 186, insert “the” before “Normative”

- lines 261 and 265, “a youth” does not make any sense as it suggests that the authors are describing a case study of a single individual

- line 272, unclear what the reference is as it reads “44-44”.  I think it should be 54?

- line 296, should read “research has shown”

Author Response

Response to Reviewer 3 Comments

Point 1. The authors categorize respondents into categories based on the “IAT” but then treat the scale as continuous for their models.  The two methods should not be done in the same paper.  Tables 1 and 2 should be a single correlation matrix with age and sex included within the matrix.  The authors also do a very poor job in explaining their “serial and separate mediation” models.  A single straightforward regression predicting internet addiction would have been much better for this manuscript.

Response 1. We thank the reviewer for these useful comments. We have implemented our plan of analyses. Specifically, we have maintained chi-square analyses, but reporting the percentages within IA-group and within sex, because they may be informative for epidemiological studies (See the new Table 2). Moreover, although we could have considered the IAT as a continuous variable from the beginning, we have chosen to preliminarily explored the associations between psychosocial and demographical variables and the different groups of IA, as suggested by previous studies. See, for example, the study by:

  • Arslan, G. (2017). Psychological maltreatment, forgiveness, mindfulness, and internet addiction among young adults: A study of mediation effect. Computers in Human Behavior72, 57-66.
  • Lim, J.-A., Gwak, A. R., Park, S. M., Kwon, J.-G., Lee, J.-Y., Jung, H. Y., … Choi, J.-S. (2015). Are Adolescents with Internet Addiction Prone to Aggressive Behavior? The Mediating Effect of Clinical Comorbidities on the Predictability of Aggression in Adolescents with Internet Addiction. Cyberpsychology, Behavior, and Social Networking, 18(5), 260–267.doi:10.1089/cyber.2014.0568
  • Lyvers, M., Karantonis, J., Edwards, M. S., & Thorberg, F. A. (2016). Traits associated with internet addiction in young adults: Potential risk factors. Addictive behaviors reports3, 56-60.

However, as suggested by the reviewer, we have chosen to modify our plan of analyses. Specifically, after conducting descriptive analyses and chi-square analyses, we have directly analyzed the possible relationship between psychosocial and demographical variables and IA group with multinomial regression analyses. We have considered as dependent variable the three-group of IA whereas sex, age, living setup, and all considered dimension of FAD, BIS-11, and ASR as independent variables. Moreover, to examine possible sex and age moderations, we created a series of interaction terms for inclusion in the final model. (see lines 177-186 of the statistical analyses, lines 217-226 of the results, the new Table 3, and lines 285,286, 295-298 of the Discussion).  Finally, from a more accurate examination of the literature we thought it more useful to redefine our conceptual model. In fact, studies in the field, rather than suggesting a predictive effect of depression on impulsivity, show the presence of associations and are mostly correlational studies. Consequently, the current literature does not support the existence of a possible sequential mediation model, and we have modified the relative sentences throughout the manuscript and our aim (see line 115), in order to verify the possible presence of a parallel mediation model. Coherently, we have modify the statistical analyses subsection (see line 188), the results section (lines 237-253), the Figure showing results of parallel mediation model (see new Figure 1), the discussion of the relative results (See line 323-330), and the conclusions of the study (see lines 368-373).

Point 2: On page 3:

*make sure you use quotation marks around the test item examples

- as suggested by the reviewer, we have added quotation marks around all the test item examples (See line 140-144, and 165-167)

*explain what is meant when you say, “each family member attests...”.  The text above suggests that only students completed the scale and not their family members

- we thank the reviewer for this comments. There was an error that we have modifies, specifying that only young adults attests the FAD (See line  144)

*specify which scale had an alpha of .72 and which was .92 (line 125)

- we have specified the alpha of all scales (See lines 148-149).

*state the alpha value for the ASR for this sample

- we have added alpha valued for the ASR for our sample (see line 160)

* state the alpha value for the BIS-11 for this sample

-  we have added alpha valued for the BIS-11 for our sample (see line 169,170)

Point 3. On page 4, note that reliability values are not qualitative as claimed on line 146, but are indeed quantitative

Response 3. As suggested by the reviewer, we have removed the word qualitative in relation to statistical analyses conducted.

Point 4. In addition, the authors claim on page 6, line 222, that their sample is from “the general population”.  This claim is incorrect as the sample consists of university students.

Response 4. We thank the reviewer for this pertinent comment. As suggested by the reviewer, we have modify the title in “Internet addiction among young adult university students: the complex interplay between family functioning, impulsivity, depression and anxiety” and described our sample in terms of university students (See lines 16,75,108, 129, 200, 295,334,350, and 354)

Point 5. More minor points:

* insert the reference for the IAT in the Abstract number 54

- we have added the reference for the IAT [63] in the Abstract (see line 19)

* page 2 – “at this regard” is an awkward phrase and the rest of the sentence needs rewriting for clarity – in this field

- we have rewriting the sentence (see lines 60-62)

* line 53, remove “the” before “research”

- we have removed “the” before “research”

* lines 65 and 72, change “showed” to “shown”

- we have changed “showed” in “shown” (see lines 68, 80)

* line 69, remove “the” before “relationship”

- we have removed “the” before “relationship” (see line 72)

* line 72, you need to add what variable was “associated”; associated with....IA? finish the clause

- we have added that variables were associated “with IA” (see line 80)

* line 85, insert “the” before “influence”

- we have inserted “the” before “influence” (see line 102)

* line 110, change to “maximum”

- we have changed “maximus” to “maximum” (see line 132)

* line 111, change to “score” (singular)

- we have changed “scores” to “score”

* line 120, clarify that the item is negatively keyed (“We don’t...”)

- we have modified the item (see line 142) and specified that higher scores of FAD indicate worse levels of family functioning (see line 146)

* line 147, change to read “was used”

- we have changed to “was used” (see line 175)

* line 163, should be “did not complete”

- we have modified the passage in “did not complete” (see line 196)

* line 167, delete “The” and write out the number

- we have delete “the” before the number (see line 200)

* line 168, delete “are” before “married”

- we have deleted “are” before “married” (see line 201)

* line 173, explicitly state that none of the sample met the criteria of “severely addicted”

- We have explained from the description of results that we did not find any young adults into the severity category (See line 207)  

* line 186, insert “the” before “Normative”

- we have  modified our analyses and results. So, this passage is no longer there

* lines 261 and 265, “a youth” does not make any sense as it suggests that the authors are describing a case study of a single individual

- we have modified “a youth” to “young adults” (see lines 311, 316)

* line 272, unclear what the reference is as it reads “44-44”.  I think it should be 54?

- thank you for this comment. We have modified the references. (See line 323)

* line 296, should read “research has shown”

-  we have  modified our results and discussion. So, this passage is no longer there

Reviewer 4 Report

This paper presents results of a study that aimed "to assess the relationship between youths' Internet addiction and youths' sex, the perception of their family functioning, impulsivity level, and depressive and anxiety symptoms".

Suggestions and questions (answers can/should be used to improve the paper):
1. Consider "...official diagnostic criteria have not yet been identified [20]". Reference 20 is too old to substantiate the claim.
2. Why is TableS1 presented as supplementary material? It could be in the manuscript directly.
3. What are the contributions of the paper? Two points: a) What are the new [main] findings? b) What are they useful for? These pieces of information should be explicit in the manuscript.
4. What are the future work (or recommendations for further investigation)? This information could be added in the conclusion section.
5. What was the hypothesis of the study? This information is missing. What was not expected? My motivation for these questions is related to some sentences of the discussion section, such as "As expected" or "results did not confirm what was expected".

Specific comments:
- lines 147-148: "The IAT cut-off of Italian validation [55] will use to divided the total sample in groups" -> will be used? to divide?
- Contractions should be avoided (e.g., it's, don't).
- Figure 1 is in low quality.
...
The paper is very well written, but a careful review is required. There are some typos / grammar errors.

Author Response

Response to Reviewer 4 Comments

Point 1. Consider "...official diagnostic criteria have not yet been identified [20]". Reference 20 is too old to substantiate the claim.

Response 1. We thank the reviewer for this comment. As suggested by the reviewer, we have changed the references with a more recent (see new reference [22]: Kurniasanti, K.S., Assandi, P., Ismail, R.I., Nasrun, M.W.S., Wiguna, T. Internet addiction: a new addiction?. Med J Indones. 2019, 28, 82-91. doi: 10.13181/mji.v28i1.2752)

Point 2. Why is TableS1 presented as supplementary material? It could be in the manuscript directly.

Response 2. We thank the reviewer for this comment. However, we have modified our statistical analyses, and MANOVA has not been conducted anymore. However, we have reported descriptive mean of psychosocial variables, both by IA-group and sex (See new Table 1, page 5). Moreover, after conducting descriptive analyses and chi-square analyses, we have directly analyzed the possible relationship between psychosocial and demographical variables and IA group with multinomial regression analyses. We have considered as dependent variable the three-group of IA whereas sex, age, living setup, and all considered dimension of FAD, BIS-11, and ASR as independent variables. Moreover, to examine possible sex and age moderations, we created a series of interaction terms for inclusion in the final model. (see lines 177-186 of the statistical analyses, lines 217-226 of the results, the new Table 3, and lines 285,286, 295-298 of the Discussion).

Point 3.  What are the contributions of the paper? Two points: a) What are the new [main] findings? b) What are they useful for? These pieces of information should be explicit in the manuscript.

Response 3. We thank the reviewer for this useful comment. We have made the main results of our study more explicit. Specifically, our findings may suggested a key role played by youths’ impulsivity level (i.e., attentional impulsivity) in the relationship between family functioning and IA, for both males and females, providing additional preliminary support for the importance of the planning of family-focused prevention programs for all young adults at risk of IA. Moreover, based on our preliminary findings, prevention focused on early detection and intervention in impulsivity and depression problems seems to be also necessary, for both males and females (See lines 351-354 of the Discussion, and lines 373-376 of the Conclusion)

Point 4. What are the future work (or recommendations for further investigation)? This information could be added in the conclusion section.

Response 4. We thank the reviewer for this pertinent comment. Based on our preliminary results, further studies should replicate our findings  with larger sample sizes of young adults in order to generalize our preliminary findings. Moreover, given the cross-sectional nature of the study, further longitudinal studies exploring the IA trajectories could be useful to identify more accurate risk factors for IA, and to support the casual connection between young adults’ family functioning, impulsivity, depression problems, and IA suggested by our preliminary findings (See lines 351-354 of the Discussion, and lines 373-376 of the Conclusion)

Point 5.  What was the hypothesis of the study? This information is missing. What was not expected? My motivation for these questions is related to some sentences of the discussion section, such as "As expected" or "results did not confirm what was expected".

Response 5. As suggested by the reviewer, we have added the hypothesis of our study in relation to each aims (see lines 111-118)

Point 6.Specific comments:

*lines 147-148: "The IAT cut-off of Italian validation [55] will use to divided the total sample in groups" -> will be used? to divide?

- we changed “will be used to divide” to “was used to divide” (see line 175)

* Contractions should be avoided (e.g., it's, don't).

- we have deleted all contractions throughout the manuscript

* Figure 1 is in low quality.

- we have implement the quality of new Figure 1

* The paper is very well written, but a careful review is required. There are some typos / grammar errors.

- we have checked and corrected typos/grammar errors throughout the manuscript

Thank you again for your kind attention.

Best regards

The authors.

Round 2

Reviewer 1 Report

I would like to thank the authors for their careful consideration to my suggestions. I wish the authors the best with their work. 

Reviewer 3 Report

Thank you for addressing my comments.